# Optimization Co-Culture of *Monascus purpureus* and *Saccharomyces cerevisiae* on Selenium-Enriched *Lentinus edodes* for Increased Monacolin K Production

**DOI:** 10.3390/jof10070503

**Published:** 2024-07-20

**Authors:** Yi He, Huafa Lai, Jinxiao Liang, Lu Cheng, Lixia He, Haolin Wang, Qingqing Teng, Wenjing Cai, Rui Wang, Lisha Zhu, Zhengbin Pang, Dafu Zhang, Xingxing Dong, Chao Gao

**Affiliations:** 1National R&D Center for Se-Rich Agricultural Products Processing, Hubei Engineering Research Center for Deep Processing of Green Se-Rich Agricultural Products, School of Modern Industry for Selenium Science and Engineering, Wuhan Polytechnic University, Wuhan 430023, China; yi.he@whpu.edu.cn (Y.H.); laihuafa123@126.com (H.L.); 15502788963@163.com (J.L.); 18371363909@163.com (L.C.); 18883246381@163.com (L.H.); tqq15107141417@163.com (Q.T.); rainbow258456@163.com (W.C.); 15377194169@163.com (R.W.); zhulisha199888@126.com (L.Z.); p2862085820@163.com (Z.P.); dongxingxinghg@whpu.edu.cn (X.D.); 2Key Laboratory for Deep Processing of Major Grain and Oil, Hubei Key Laboratory for Processing and Transformation of Agricultural Products, Ministry of Education, School of Food Science and Engineering, Wuhan Polytechnic University, Wuhan 430023, China; 3Suixian Public Inspection and Testing Center, Suizhou 441300, China; wangilous@163.com; 4Hubei Hongyang Ecological Technology Co., Ltd., Suizhou 441300, China; 15357929614@126.com; 5Hubei Hetai Food Co., Ltd., Suizhou 441300, China; 6Hubei Zhongxing Food Co., Ltd., Suizhou 441300, China

**Keywords:** *Monascus purpureus*, fermentation, selenium-enriched *Lentinus edodes*, monacolin K, antioxidant

## Abstract

Selenium-enriched *Lentinus edodes* (SL) is a kind of edible fungi rich in organic selenium and nutrients. *Monascus purpureus* with high monacolin K (MK) production and *Saccharomyces cerevisiae* were selected as the fermentation strains. A single-factor experiment and response surface methodology were conducted to optimize the production conditions for MK with higher contents from selenium-enriched *Lentinus edodes* fermentation (SLF). Furthermore, we investigated the nutritional components, antioxidant capacities, and volatile organic compounds (VOCs) of SLF. The MK content in the fermentation was 2.42 mg/g under optimal fermentation conditions. The organic selenium content of SLF was 7.22 mg/kg, accounting for 98% of the total selenium content. Moreover, the contents of total sugars, proteins, amino acids, reducing sugars, crude fiber, fat, and ash in SLF were increased by 9%, 23%, 23%, 94%, 38%, 44%, and 25%, respectively. The antioxidant test results demonstrated that 1.0 mg/mL of SLF exhibited scavenging capacities of 40%, 70%, and 79% for DPPH, ABTS, and hydroxyl radicals, respectively. Using gas chromatography–ion mobility spectrometry technology, 34 unique VOCs were identified in SLF, with esters, alcohols, and ketones being the main components of its aroma. This study showed that fungal fermentation provides a theoretical reference for enhancing the nutritional value of SL.

## 1. Introduction

Selenium (Se) is a vital trace element for both humans and animals. It serves as an essential nutrient and plays a crucial role in the regulation of Se-dependent enzymes in cells. In addition, Se possesses antitumor, antiaging, antioxidant, and cardiovascular and cerebrovascular disease prevention effects, while enhancing the immune system [1]. *Lentinula edodes* is a highly regarded fungus frequently consumed in China. *Lentinus edodes* shares a common origin with food and medicine and possesses significant nutritional, medicinal, and health benefits. Notably, *Lentinus edodes* is high in protein and low in fat and contains polysaccharides, various amino acids, and vitamins. It has special antitumor effects and helps prevent and treat cardiovascular diseases, as well as effectively lowering blood lipid levels, strengthening the body’s immune system, and ultimately, improving health. Se incorporation into reactive macromolecules during cellular material metabolism enables the edible fungi mycelium in converting inorganic Se from substrates into selenoproteins, selenopolysaccharides, and other organic forms, characterized by low toxicity and high bioavailability [2]. Furthermore, Kaleta et al. [3] demonstrated the potential of shiitake mushroom mycelium as a novel source of immunomodulatory drugs, chemopreventive agents, and food supplements. Moreover, evidence indicates that Se enrichment positively influences the antioxidant activity of mycelial extracts.

The genus *Monascus* is classified in the class Ascomycetes and the family Monascaceae [4]. *Monascus* spp., particularly *M. purpureus* and *M. ruber*, as filamentous fungi with both medicinal and edible properties, are extensively utilized in industries such as medicine, healthful wine production, fermented bean curd, and the production of fermented foods. They serve as pivotal ingredients in the fermentation of products, such as red yeast rice. Owing to their varied health-promoting metabolites, such as lovastatin (monacolin K, MK), pigments, and γ-aminobutyric acid, *Monascus*-related foods have gained global recognition. MK is a secondary metabolite of *Monascus* strains. In the US, MK was the inaugural hypocholesterolemic medication to be authorized by the FDA. This drug class is globally acknowledged as the preferred medication for managing hyperlipidemia and averting and treating atherosclerosis, coronary heart disease, and cerebrovascular disease. The diverse array of beneficial compounds has garnered attention from the scientific community [5]. Among various microorganisms, *Saccharomyces cerevisiae* (*S. cerevisiae*) is a highly effective biocontrol agent because of its biology and nontoxic characteristics. The metabolites generated by the growth of *S. cerevisiae* can enhance *Monascus* pigments. Hydrolytic enzymes, such as chitinase, produced by *S. cerevisiae*, contribute to damaging the cell wall of *Monascus*, thereby enhancing pigment production [4]. Shi et al. [6] observed that cocultivation with *S. cerevisiae* markedly upregulated the expression levels of genes associated with MK production (*mokA*-*mokI*), with particular emphasis on *mokE*, *mokF*, and *mokG*. Fermented foods are significant components of cultures and societies worldwide. They are recognized for their nutritional and health advantages, with research indicating their substantial impact on managing particular chronic illnesses [7].

Bio-fermentation technology is widely employed in agriculture and has emerged as a potent tool for advancing the development of novel sustainable food products with enhanced nutritional value [8]. He et al. [9] reported that the fermentation quality and nutritional profile of rice straw silage were enhanced by the mixing of *Monascus*-fermented *Moringa oleifera* leaves. This intervention resulted in a reduction in pH, butyric acid, ammonia nitrogen, and fiber content, while simultaneously increasing the protein content of the rice straw silage. Microbial fermentation has demonstrated the ability to improve the antioxidant potential of peach juice and plum by-products, as well as the anti-inflammatory and antioxidant characteristics of hydroponically cultivated ginseng and bee pollen [10]. Common fermentation methods for *Monascus* in manufacturing MK comprise solid-state fermentation (SSF) and submerged fermentation. The fermentation of diverse food substrates by *Monascus* spp. increases their antioxidant properties [11]. Several studies have demonstrated a link between flavor and metabolites. For instance, a decrease in hypoxanthine levels can diminish the bitter compounds in *Crassostrea gigas*, and the conversion of fatty acids into volatiles can also boost meat flavor. To reveal the mechanisms governing the formation of meat quality and flavor characteristics in Peking greasy chicken, metabolomics was used. Moreover, gas chromatography–ion mobility spectrometry (GC–IMS ), an inventive technique combining GC and IMS, was employed to promptly identify volatile organic compounds (VOCs) [12].

Therefore, this study primarily aimed to optimize the fermentation process for MK production and conduct analyses on the nutritional components, antioxidant capacity, and volatile flavor compounds of selenium-enriched *Lentinus edodes* fermentation products. Results will be important for understanding the added value of selenium-enriched *Lentinus edodes* and functional product development.

## 2. Materials and Methods

### 2.1. Materials

*Saccharomyces cerevisiae* (*S. cerevisiae*) CICC 1575 and *Monascus purpureus* (*M. purpureus*) CICC 5046 were purchased from the China Center of Industrial Culture Collection, whereas *Monascus ruber* (*M. ruber*) HY1 was screened and stored in our laboratory. Selenium-enriched *Lentinus edodes* (SL) were obtained from Shenxiang 1513 *Lentinus edodes* grown with selenium nutrient solution in Xinji Village, Sanligang Town, Suizhou City, Suizhou County, Hubei Province, which were collected, lyophilized, and ground into powder at 60 mesh and sieved. SL powder (2 g) was placed into a 250 mL conical flask, followed by the addition of glucose and distilled water in a specific ratio. The mixture was then sterilized at 115 °C for 20 min to prepare the required fermentation medium. Amino acid standards were procured from Wuhan Xinshen Test Co., Ltd. (Wuhan, China). Glucose, HCl, HNO_3_, NaOH, H_3_PO_4_, H_2_SO_4_, and KBH_4_ were also obtained from Wuhan Xinshen Test Co., Ltd. (Wuhan, China). A 1000 μg/mL Se standard solution was purchased from Guoyao Group Chemical Reagent Co., Ltd. (Wuhan, China).

### 2.2. Determination of MK, Biomass, and Color Values for Strain Screening

To select the *Monascus* strain that exhibits remarkable advantages and potential in lipid-lowering, food coloring, and industrial applications, we conducted assessments on the MK content, biomass, and color value of the freeze-dried powder derived from *M. purpureus* and *M. ruber*. *M. purpureus* and *M. ruber* were inoculated onto Potato Dextrose Agar (PDA) plates overlaid with cellophane. The cultures were collected after 5, 10, 15, 20, and 25 days, freeze-dried, and stored in a desiccator for subsequent testing. The MK was evaluated using high-performance liquid chromatography (HPLC), following the method outlined by Wang et al. [13], with minor modifications. The sample was accurately weighed to 0.50 g, and 5 mL of acetonitrile was added. Ultrasonic extraction was performed after shaking (130 revolutions per min (rpm)) at 50 °C in a water bath for 1 h. After treatment, the solutions were centrifuged at a rate of 8000 rpm for 10 min. Subsequently, the supernatant was filtered using a micropore film with a 0.22 μm pore size. The MK concentration in SLF was determined and analyzed using HPLC. The HPLC system parameters (Agilent, Santa Clara, CA, USA) were as follows: a C18 chromatographic column (5 µm, 4.6 × 250 mm) operated at 35 °C, the mobile phase consisted of a mixture of acetonitrile–0.01% H_3_PO_4_ (3:2, *V*:*V*), the flow rate was set at 1.0 mL/min, the injection volume was 10 µL, and detection occurred at a wavelength of 238 nm. The biomass and color values of the samples were assessed using the methodology outlined by Chatepa et al. [14], with minor modifications.

### 2.3. Experiment Design for a Single Factor and Determination

The pretest indicated that the fermentation temperature, medium ratio, fermentation time, spore concentration, and inoculum ratio have effects on the yield of MK. Consequently, these five factors were selected for the experimental investigation. Different concentrations of *M. purpureus* and *S. cerevisiae* spores (10^5^, 10^6^, 10^7^, 10^8^, and 10^9^ spores/mL), fermentation temperatures (24 °C, 28 °C, 32 °C, 36 °C, and 40 °C), media ratios (media ratios of SL, glucose, and distilled water were set as 1:1:5, 1:2:10, 1:3:15, 1:4:20, and 1:5:25), fermentation time (12, 16, 20, 24, and 28 days), and inoculum ratio (*M. purpureus:S. cerevisiae* = 1:3, 1:2, 1:1, 2:1, and 3:1) were investigated as single factors to elucidate their impacts on fermentation and MK production by SL, while maintaining other conditions constant [13,15].

### 2.4. Plackett–Burman Experiment Design

Following the outcomes of the one-way analysis, it was established that the fermentation temperature, medium ratio, fermentation time, spore concentration, and inoculum ratio significantly influenced MK production in SLF. Subsequently, to identify the three most influential factors affecting MK production, the Plackett–Burman experimental design was employed [16].

### 2.5. Response Surface Design

This study aimed to optimize the fermentation conditions for achieving the maximum MK yield. Adhering to the Box–Behnken design principles, the MK content was selected as the response variable, with fermentation time, fermentation temperature, and medium ratio selected as independent variables. To optimize the fermentation process, a three-factor, three-level Box–Behnken design was employed using Design-Expert 8.0 software (StatEase Inc., Minneapolis, MN, USA), followed by verification.

### 2.6. Nutrient Composition Analysis

#### 2.6.1. Determination of Se Content

The samples underwent ultrasonography-assisted acid digestion, following the procedure outlined by Rebellato et al. [17], with minor modifications. Approximately 0.200 g of the sample was weighed into 50 mL graduated tubes. Subsequently, 7 mL of HNO_3_ was added, and digestion was performed at 170 °C. Once cooled, the solution was mixed with 12 mol/L of HCl in a 1:1 (*V*:*V*) ratio and agitated in an Intelligent Temperature Control Electric Heater (G-400, Shanghai Yiyao Instrument Technology Development Co., Ltd., Shanghai, China) at 180 °C for 30 min. The Se concentration was determined using an atomic fluorescence spectrometer (AFS8530, Beijing Haiguang Instrument Co., Ltd., Beijing, China). The samples were placed in a 50 mL graduated stoppered test tube, and 20 mL of 50% hydrochloric acid solution was added. After thorough mixing, the mixture was placed in a 70 °C constant-temperature water bath and shaken at 150 rpm for 2 h. Following this, it was cooled to room temperature and filtered through an absorbent cotton ball. The cotton ball was rinsed with a small amount of 50% hydrochloric acid solution, and the filtrate was collected and diluted to 50 mL. The filtrate was then poured into a separatory funnel, where 5 mL of cyclohexane was added for extraction. After allowing the layers to separate, the aqueous phase was collected. Subsequently, 25 mL of the aqueous phase was transferred to a 50 mL graduated stoppered test tube and heated in a boiling water bath for 20 min. After cooling to room temperature, 2.5 mL of 10% potassium ferricyanide solution and 3 drops of n-octanol were added, and the volume was adjusted to 50 mL with water, ready for measurement. The inorganic Se concentration was determined using an atomic fluorescence spectrometer. Hydrochloric acid (10%) was used as a carrier stream and potassium borohydride solution (10 g/L) as a reducing agent. The operating parameters for the atomic fluorescence spectrometer were set at a negative high voltage of 300 V, a lamp current of 80 mA, a carrier gas flow rate of 300 mL/min, a shield gas flow rate of 800 mL/min, a delay time of 6 s, and a reading time of 18 s. The determination of the total organic selenium concentration was achieved by deducting the level of inorganic selenium from the overall selenium concentration [18].

#### 2.6.2. Nutritional Composition

According to the “Official Methods of Analysis of AOAC International” (AOAC, 2012), total sugar (AOAC 984.26), reducing sugar (AOAC 982.14), ash (AOAC 942.05), crude fiber (AOAC 962.09), and fat (AOAC 963.15) were determined gravimetrically [19]. The Kjeldahl method was used to calculate total nitrogen (AOAC992.15), and the result was multiplied by 4.38 (a conversion factor specifically used for SL) for protein estimation [19]. The amino acid composition of the proteins of the samples was determined using a Hitachi L-8900 automatic amino acid analyzer (Tokyo, Japan) [20].

### 2.7. Antioxidant Activity

#### 2.7.1. Preparation of Samples for Antioxidant Determination

The active components of the samples were extracted using ultrasonic extraction, following the method described by Chatepa et al. [14], with minor modifications. The samples were quickly ground using a grinder, sieved through a 100-mesh sieve, and weighed to 50 g in a conical flask. Subsequently, the samples were extracted with the following five organic solvents: methanol, ethanol, acetone, ethyl acetate, and petroleum ether. The extraction was performed with a liquid-to-material ratio of 5:1, at an ultrasonic temperature of 30 °C, for a 2 h ultrasonic duration, with a 100 Hz ultrasonic power. The extracted liquids were combined, filtered, and concentrated to dryness using a rotary evaporator. Solutions with 0.05, 0.1, 0.25, 0.5, and 1.0 mg/mL concentrations were prepared for antioxidant activity determination. Ascorbic acid (Vc) served as the positive control in all experiments.

#### 2.7.2. 2,2-Diphenyl-1-picrylhydrazyl (DPPH) Radical Scavenging Assay

A 0.2 mmol/L ethanolic solution of DPPH was prepared following a modified method described by Zhang et al. [21]. To create the reaction mixture, 2 mL of the DPPH ethanolic solution and 2 mL of the sample solution were combined. The mixture was subsequently shaken and left in a dark room at room temperature (25 °C) for 30 min. In parallel, blank groups, in which samples were replaced with distilled water, and control groups, in which the DPPH solution was substituted with absolute ethanol, were prepared. The DPPH scavenging rate was determined by analyzing the absorbance values recorded at 517 nm:Scavenging rate (%) = (1 − [A_2_ − A_1_]/A_0_) × 100
where A_0_ refers to the absorbance of the blank groups, A_1_ refers to the absorbance of the control groups, and A_2_ refers to the absorbance of the samples.

#### 2.7.3. 2,2′-Azino-bis-3-ethylbenzothiazoline-6-sulfonic Acid (ABTS) Radical Scavenging Assay

Following the method outlined by Zhang et al. [10], some modifications were introduced for the preparation of ABTS radicals. The ABTS stock solution was formulated by mixing 1 mL of ABTS solution (7.4 mmol/L) with 1 mL of potassium persulfate solution (2.6 mmol/L) and allowing it to incubate in darkness for 12 h. Subsequently, 800 µL of the ABTS stock solution, which was diluted with ethanol, was mixed with 200 µL of the sample. The control groups, in which samples were replaced with ethanol under otherwise unchanged conditions, were included. After incubation in the dark for 6 min, the final absorbance value was measured at 735 nm:Scavenging rate (%) = (1 − [A_2_ − A_1_]/A_0_) × 100
where A_0_ refers to the absorbance of the blank groups, A_1_ refers to the absorbance of the control groups, and A_2_ refers to the absorbance of the samples.

### 2.7.4. Hydroxyl Radical Scavenging Assay

The hydroxyl radical scavenging capacity was assessed following the procedure outlined by Liu et al. [22]. In summary, a 1 mL sample solution was combined with 1 mL of FeSO_4_ (9 mmol/L) and 1 mL of H_2_O_2_ (8.8 mmol/L) at 37 °C for 10 min. The absorbance of the resulting supernatant was subsequently measured at 550 nm:Scavenging rate (%) = (1 − [A_2_ − A_1_]/A_0_) × 100
where A_0_ denotes the absorbance of the blank groups, A_1_ represents the absorbance of the control groups, and A_2_ signifies the absorbance of the samples.

## 2.8. Volatile Compounds Analysis

To detect VOCs in the SL and SLF, a FlavourSpec^®^ Flavor Analyzer (FlavourSpec^®^, Dortmund, Germany) was used. The samples were precisely weighed at 1.0 g per sample and then transferred to 20 mL headspace flasks. The flasks were sealed with PTFE-coated lids and left for measurement. The following were the headspace incubation parameters: Samples were incubated at 80 °C for 15 min at an incubation speed of 500 rpm. The injection needle temperature was maintained at 85 °C, with an injection volume of 0.5 mL applied. The non-shunt mode was utilized, with injection conducted at a rate of 150 µL/min, with high-purity nitrogen used to push and clean the headspace injection needle. The cleaning process lasted for 15 min. The GC–IMS study conditions were set at 45 °C for the migration spectrum temperature, with N_2_ gas migration at a rate of 150 mL/min and purity exceeding 99.999%. The positive-ion mode was employed for ionization, with β-rays (tritium, 3H) applied for radiation treatment [12].

## 2.9. Statistic Evaluation

Experimental data were analyzed using IBM SPSS Statistics 20.0, Microsoft Office Excel 2010, and the GC–IMS autonomous software for data collection, specifically the LAV software (version 2.0.0) from G.A.S. in Dortmund, Germany. The results of each experiment were conducted in triplicate and are presented as mean values ± standard deviations. To discern variations among different groups, independent-samples *t*-tests and one-way analysis of variance (ANOVA) were utilized, with Tukey’s post hoc test applied. A significance level of *p* < 0.05 was utilized for statistical inference.

## 3. Results and Discussion

### 3.1. Screening of the Test Strains

The accumulation of MK in *M. purpureus* began on the 5th day, with a significant increase in MK production observed after the 10th day (Figure 1a). On the 25th day, the MK yield reached 32.97 mg/g. However, as time progressed, MK production by *M. ruber* consistently remained significantly lower than that by *M. purpureus*. On the 25th day, *M. ruber* produced only 3.09 mg/g of MK. The elevated MK production capacity of *M. purpureus* contributes to the heightened tolerance of fungal cells to the secondary metabolite MK. This underscores the superior suitability of *M. purpureus* for MK production, particularly in the later stages of cultivation. It has been shown that *M. purpureus* is a potent source of compounds (MK, dimerumic acid, and gamma-aminobutyric acid) that have anti-obesity properties, which help in combating diseases [23].

As the cultivation period progressed, fungal biomass continued to increase (Figure 1b). Additionally, the fungal biomass of *M. purpureus* was significantly higher than that of *M. ruber*. The biomass of *M. purpureus* was approximately double that of *M. ruber* from the 15th to 25th day. The increase in biomass indicates more active growth and reproductive activity in *M. purpureus* and may imply an increase in some secondary metabolites.

Figure 1c shows that *M. purpureus* reached its peak pigment value on the 10th day, reaching approximately 2200 U/g. Furthermore, the pigment values remained consistently above 1400 U/g from the 5th to 20th day, significantly surpassing the pigment values of *M. ruber*. Conversely, the pigment value of *M. ruber* gradually decreased from its peak on the 5th day (1394 U/g) to approximately 247 U/g. Some studies have noted that *M. purpureus* demonstrates a superior capacity for pigment production compared with *M. ruber*, particularly with regard to monascin and ankaflavin synthesis [24].

Therefore, *M. purpureus* is a more suitable strain for subsequent fermentation. In addition, *M. purpureus* is one of the safest molds for MK production and has been used in Chinese medicine preparation.

### 3.2. Single-Factor Experimental Analysis

Various factors, such as the fermentation time (A), fermentation temperature (B), spore concentration (C), medium ratio (D), and inoculum ratio (E), contribute to the growth rate of *M. purpureus* and *S. cerevisiae*, which in turn affects the efficiency of metabolite synthesis. This experiment examined the influence of these factors on MK synthesis. During the fermentation process (Figure 2a), the MK content steadily increased, rapidly increasing from 0.11 ± 0.03 to 1.74 ± 0.08 mg/g. However, between the 16th and 20th days, the rate of increase slowed down and finally remained at approximately 2 mg/g. During the experiment, the fermentation time increased, and the amount of material at the bottom of the container decreased. This is believed to be caused by the late stage of fermentation, leading to the consumption of substrate nutrients, inhibiting fungal growth, and consequently affecting the MK content [25]. Based on these results, 20 days is the most suitable fermentation time.

At a fermentation temperature of 24 °C, the fungal growth rate was sluggish, and the red mold exhibited a lighter color, accompanied by a markedly low MK content of only 0.16 ± 0.01 mg/g (Figure 2b). At a fermentation temperature of 28 °C, fungal fermentation proceeded smoothly, *M. purpureus* displayed a dark red color, and the MK content reached the highest value of 1.81 ± 0.05 mg/g. When the fermentation temperature was 40 °C, the fungal growth rate was diminished, thereby leading to a concomitant reduction in secondary metabolite production. Temperature exerted a direct influence on the biological activity, metabolic rate, and production of metabolites by microorganisms in the context of food fermentation [23]. The findings suggest that the MK content in SLF was highest at 28 °C. Studies have demonstrated that the strains *M. purpureus* MTCC369 and *M. ruber* MTCC 1880 exhibited the highest MK yield when cultured at a constant temperature of 29.46 °C, which aligns with the findings of our experimental study [15]. Therefore, the best fermentation temperature was selected as 28 °C.

MK concentration gradually increased from 0.70 ± 0.04 to 1.44 ± 0.04 mg/g as the spore concentration increased from 10^5^ to 10^7^ spores/mL (Figure 2c). However, the MK content gradually decreased from 1.44 ± 0.04 to 0.28 ± 0.03 mg/g when the spore concentration exceeded 10^7^ spores/mL. Therefore, the results suggested that excessive inoculation negatively affected the efficiency of *M. purpureus* and *S. cerevisiae* fermentation and the production of secondary metabolites in the later stages when the concentration exceeded the optimal value. Similar to previous studies, the spore count for achieving the highest MK production was 5 × 10^7^ spores/mL [23]. Thus, the spore concentration of 10^7^ spores/mL was considered appropriate.

It is noteworthy that when the inoculum ratio (*M. purpureus*:*M. ruber*) was set from 2:1 to 1:1, the maximum increase in MK yield was observed, with the highest yield reaching 1.98 ± 0.03 mg/g (Figure 2d). This may be because variations in the inoculum ratio affect the MK content of the product and the overall fermentation efficiency [26]. If the inoculum ratio was above or below 1:1, both fungal growth and metabolite formation were inhibited, thereby reducing the accumulation of the desired product, MK. As a result, the inoculum ratio of 1:1 was used in the following experiment.

Water content played a pivotal role in regulating fermentation when SL was employed as the substrate matrix. The results demonstrated that the optimal MK content was attained when the SL–glucose–distilled water ratio was 1:3:15, reaching 1.81 ± 0.05 mg/g (Figure 2e). Beyond this ratio, the MK content gradually declined, reaching a minimum concentration of 0.39 ± 0.04 mg/g. As fermentation progressed, the medium lost a significant amount of water, gradually drying out the substrate. This hindered the growth and metabolism of *S. cerevisiae*, thereby resulting in reduced metabolite production, including MK [25]. Furthermore, studies have shown that *S. cerevisiae* can stimulate the production of secondary metabolites in *M. purpureus* [4]. However, when nutrients in the medium become scarce and the environment turns unfavorable, the growth rate of yeast may slow down or even cease, accompanied by a corresponding decrease in metabolic activities, ultimately resulting in reduced MK synthesis. Hence, the medium ratio of 1:3:15 was considered optimal.

Overall, the MK content approached the peak value when the fermentation time, fermentation temperature, concentration, inoculum volume ratio, and medium ratio reached 28 °C, 20 days, 10^7^ spores/mL, 1:1, and 1:3:15, respectively.

### 3.3. Fermentation Process Main Effects

#### 3.3.1. Plackett–Burman Design Test and Result Analysis

Initially, to evaluate the influence of these factors on MK production, a series of 12 experiments was performed using the Plackett–Burman design (Table 1).

#### 3.3.2. Pareto Chart

The Pareto chart, created using Design-Expert 8.0 software, emerged as an invaluable instrument for discerning the principal factor impacting the MK yield of fermentation, as depicted in Figure 3. The factor with the highest standardized effect makes the most significant contribution, whereas the factor closer to 0 indicates a minor contribution. Positive and negative effects represent desired and undesirable outcomes, respectively. The results suggested a positive influence of fermentation temperature and time on MK production, indicating the necessity to increase both temperature and time. Conversely, the medium ratio exhibited a negative effect, highlighting the need to reduce the distilled water content. Fermentation temperature (B) was the factor with the highest value of the standardized effect. The other two significant factors were medium ratio (E) and fermentation time (A). All three factors were selected to be optimized using the Box–Behnken design.

### 3.4. Response Surface Analysis

#### 3.4.1. Response Surface Test and Results

To determine the optimal combination of variables and response patterns, a Box–Behnken design encompassing three variables was employed. The levels of the three variables for the effect of fermentation time (A), fermentation temperature (B), and medium ratio (C) are presented in Table 2. To determine the effect of the MK yield, a total of 17 experiments were conducted separately.

#### 3.4.2. Model Fitting

The regression model derived from Design-Expert 8.0 software for fermentation time (A) and fermentation temperature (B) exhibited notably superior fitness compared to the model for medium ratio (C). The regression models for variables A, B, and C are delineated in the table below, with the corresponding linear regression equations provided as follows:Y = 2.12 + 0.47x_1_ + 0.36x_2_ − 0.12x_3_ + 0.27x_1_x_2_ − 0.055x_1_x_3_ + 0.029x_2_x_3_ − 0.62x_1_^2^ − 0.41x_2_^2^ − 0.61x_3_^2^

The ANOVA results for this model are presented in Table 3. The R^2^ values (0.9913) and adjusted R^2^ (0.9801) close to 1, along with a model *p*-value of <0.01 and nonsignificant lack-of-fit term, indicate the reasonability and suitability of the model for predicting MK yield. The model, along with the three quadratic terms (A^2^, B^2^, and C^2^), demonstrated statistical significance (*p* < 0.01), suggesting that the impacts of A, B, and C exerted highly significant effects on the MK yield. Table 3 shows that the interaction between fermentation time and fermentation temperature (AB) was highly significant (*p* < 0.01). Overall, it can be inferred from the *p*-values that the fermentation time and fermentation temperature had the most significant impact on the MK content of the SLF, whereas the medium ratio had the least impact. Some studies suggest that extending the fermentation time provides more time for the biosynthetic process, thereby increasing MK production. Temperature variations can affect enzyme activity and reaction rates, consequently altering the MK synthesis rate [27].

#### 3.4.3. Response Surface Analysis

Data analysis and processing were conducted using Design-Expert 8.0 software, leading to the generation of contours and 3D contour curves, as depicted in Figure 4. The correlation between fermentation time and fermentation temperature and their respective effects on the MK content in SLF are illustrated in Figure 4a,b. MK production reached its maximum level at approximately 2.23 mg/g after a fermentation time of approximately 24 days and a temperature of 32 °C during the fermentation process. MK production decreased as time or temperature increased. The correlations between media ratios and fermentation time during the fermentation process are depicted in Figure 4c,d. The MK yield increased from 0.44 to 2.21 mg/g with a 20-day fermentation time and a media ratio of 1:3:15. Figure 4e,f show that the MK content increased (2.21 mg/g) when the fermentation temperature and medium ratio reached 28 °C and 1:3:15, respectively. The results show that the contour of the Box–Behnken plot is elliptical and curved, indicating that the interaction of the two factors had a significant effect on MK content. Aligned with previous research, the optimization of *Monascus* solid-state fermentation (SSF) conditions resulted in a notable 1.6-fold increase in MK yield [13]. Grounded in practicality, Design-Expert 8.0 recommended a fermentation duration, temperature, and medium ratio of 22 days, 30 °C, and 1:3:14.5, respectively. Following these suggestions, subsequent validation experiments yielded an MK yield of 2.42 mg/g, closely resembling the predicted value of 2.35 mg/g.

### 3.5. Nutrient Composition Analysis

A comparative analysis of the nutritional components of SL and SLF revealed that the fermentation process significantly influenced the nutritional composition of food ingredients. As shown in Figure 5a, the total selenium contents of SL and SLF were 6.21 ± 0.05 and 7.33 ± 0.05 mg/kg, respectively, and the organic selenium contents were 5.78 ± 0.05 and 7.22 ± 0.07 mg/kg, respectively, which accounted for 93% and 98% of the total selenium content, respectively, indicating that organic selenium contents in SLF were enhanced. This may be attributed to the fact that during the fermentation process, both *S. cerevisiae* and *M. purpureus* employed their respective metabolic pathways and enzymatic systems to assimilate inorganic selenium (such as selenate and selenite) from SL, subsequently converting it into organic selenium forms, such as selenoproteins and selenium polysaccharides, through processes such as reduction, methylation, and selenoprotein synthesis. Some studies have proposed that, in contrast to sodium selenite, organic Se sources exhibit enhanced efficacy as antioxidant modulators, potentially boosting GSH-Px activity and Se deposition in broiler muscle tissues [28]. The increase in organic Se in the final product may positively impact the nutritional value of the food [29]. Overall, SLF exhibited a marked increase in total sugar, reducing sugar, protein, crude fiber, and fat contents (Figure 5b). Specifically, SLF exhibited higher levels of total sugar and protein, with 29.54 ± 0.30 and 22.05 ± 0.43 g/100 g, respectively, in contrast to SL samples, with 27.01 ± 0.27 and 17.88 ± 0.20 g/100 g, respectively. Furthermore, the content of reducing sugars in SLF (11.57 ± 0.12 g/100 g) was markedly higher than that in SL (5.96 ± 0.06 g/100 g), which may be because of the metabolism of sugars by the fungi during fermentation affecting the total and reducing sugar content. Some studies suggest that using fungi for the biotransformation of by-products from edible mushrooms and employing these by-products as a substrate for microbial growth through SSF can enhance their nutritional components, including elevated levels of high-biological-value minerals and proteins [30]. Regarding crude fiber content, SLF contained 9.68 ± 0.12 g/100 g, whereas SL contained 7.03 ± 0.08 g/100 g, indicating higher fiber levels that could positively impact digestive health. In addition, the fat and ash contents in SLF were 2.20 ± 0.10 and 2.20 ± 0.11 g/100 g, respectively. Ivanišová et al. [27] reported that protein, fat, reducing sugar, crude fiber, and ash contents are higher in fermented grains. Therefore, these comparisons offer consumers options for balancing diverse nutritional needs in dietary choices.

Through a comparative analysis of the amino acid content in SL and SLF, fermentation significantly impacted the amino acid contents of SL, thereby resulting in a notable increase in both total and essential amino acid contents in SLF (Table 4). This implies that, during the fermentation process, microorganisms assume a pivotal role in the proficient degradation of proteins, ultimately culminating in their conversion into amino acids. Moreover, the essential amino acid content in SLF was 8.92 g/100 g, whereas that in SLF was 5.69 g/100 g, suggesting that SLF provides higher benefits in supplying essential amino acids required by the human body. Further calculation of the essential amino acid quality score revealed a 21.7% increase in SLF (45.44%) compared with SL (35.58%). This suggests that the fermentation process allowed microorganisms (e.g., *Saccharomyces* and *Lactobacillus*) to break down complex protein structures and convert them into simpler, more easily absorbed and utilized amino acids [31]. Therefore, the fermentation process enhanced the amino acid content of the product, thereby endowing SLF with superior nutritional value. This is consistent with the findings of a recent study, which suggested that the amino acid content is more abundant in yogurt following fermentation [32].

### 3.6. Antioxidant Activity

#### 3.6.1. DPPH Radical Scavenging Assay

The DPPH radical scavenging rates of SL and SLF continued to increase when the sample concentrations ranged from 0.1 to 1.0 mg/mL, reaching maximum values of 23.80% and 39.65%, respectively (Figure 6a). Some studies have indicated that *Monascus* pigments also exhibit robust DPPH free radical scavenging activity [33], which is consistent with the findings of our study. The results suggested that the observed increase in DPPH free radical scavenging activity following fermentation can be partially attributed to red pigment production. Srivastava et al. [34] investigated the impact of fermentation on the antioxidant activity of pearl millet and observed an increase in DPPH activity from 51.43% to 97.50%.

#### 3.6.2. ABTS Radical Scavenging Assay

To assess the changes in the antioxidant activity of SL and SLF, the ABTS method was employed. Our results showed that SLF had a higher ABTS free radical scavenging activity than SL (*p* < 0.05). The ABTS free radical scavenging rates of SL and SLF at a 1.0 mg/mL concentration were approximately 54.19% and 69.98%, respectively (Figure 6b). SLF demonstrated superior ABTS scavenging capability when compared to its DPPH radical scavenging activities. Studies have shown that probiotic fermentation enhances the free radical scavenging activity of ABTS, probably because of changes in phenolic compounds [35].

#### 3.6.3. Hydroxyl Radical Scavenging Assay

The results of the hydroxyl radical experiment indicated that the hydroxyl radical scavenging rate of SLF could reach 78.91% at 1.0 mg/mL, whereas that of SL was only 19.75% (Figure 6c). Furthermore, SLF exhibited robust hydroxyl radical scavenging capacity in comparison to Vc. A plausible explanation lies in the microbial generation of enzymes during the fermentation process, which can effectively scavenge free radicals and consequently enhance the antioxidant properties of the food [36]. This finding provides valuable information for understanding the mechanism by which fermentation enhances the functional components of SL and serves as experimental evidence for improving the antioxidant properties of SL products using fermentation technology. This result is consistent with the findings of Tang et al. [37], who highlighted that fermented date palm possesses excellent antioxidant properties.

### 3.7. Volatile Organic Compounds Analysis

GC–IMS is a novel method for examining volatile flavor constituents that is rapid, efficient, and convenient, and requires no pretreatment of samples. To further explore the changes in the volatile flavor compounds of SL and SLF, the GC retention time of the volatiles and the IMS migration time were employed to qualitatively analyze the volatile components. A total of 56 VOCs were identified in SL and SLF through GC–IMS analysis (Table 5). These compounds represented 12 chemical classes, including 15 esters, 13 alcohols, 3 ketones, 5 aldehydes, 5 acids, and others.

The most prevalent VOCs noted in SLF were esters, alcohols, and ketones. To visually obtain information on the effect of volatile flavor substances before and after SL fermentation, fingerprints were generated for the ion peaks that were characterized by pattern mapping. Each bright spot in Figure 7 represents a volatile flavor compound, and the color depth of the bright spot is positively correlated with the content of volatile flavor compounds. The results shown in Figure 7 indicate a greater variety of volatile flavor substances in SLF than that in SL, with 34 compounds, consisting mainly of 11 esters, 9 alcohols, 3 acids, and 3 ketones. During fermentation, redox reactions may facilitate the production of ester compounds. These reactions may be triggered due to reactive oxygen species produced by microbial metabolism or due to the redox nature of the substrate itself. Esters included butyl formate, (Z)-3-hexenyl acetate, methyl isobutyrate, 2-methylbutyl ester, ethyl heptanoate, and methyl 3-methylbutanoate. Alcohols included 1,3-butanediol, 2-octanol, 3-methyl-2-butanol, pentan-1-ol, 3-methylbutan-1-ol, and isopropyl alcohol. This may be due to the synthesis of alcohols by *M. purpureus* and *S. cerevisiae* through their metabolic pathways during fermentation. Among the volatile organic compounds found in the fermentation products, esters, alcohols, and acids were the most abundant. Esters, alcohols, and acids constituted the highest abundance of VOCs noted in SLF. When *M. purpureus* and *S. cerevisiae* undergo fermentation, they engage in a range of biological metabolic activities, including glycolysis and the tricarboxylic acid cycle, thereby leading to the potential production of esters and alcohols [38]. Wu et al. [39] reported similar results regarding fermented chili peppers, suggesting that alcohols and esters were the primary volatile flavor compounds. Overall, the interaction of several factors, including microbial activity, the action of Se, proteolysis, and the regulation of enzyme systems, during fermentation may increase the variety of volatile flavor substances in the following fermentation of edible fungi [40]. This increase in richness may provide more complex and enriched flavors to the sensory profile of SLF.

## 4. Conclusions

To determine the optimal co-fermentation process for SL dominated by *M. purpureus* and *S. cerevisiae*, this study conducted single-factor experiments, Plackett–Burman experiments, and response surface methodology. The validated MK production yield obtained from the confirmation experiment was 2.42 mg/g. In addition, the nutritional components of SLF, including total sugar, protein, amino acids, reducing sugar, crude fiber, fat, and ash contents, exhibited significant increases following fermentation facilitated by *M. purpureus* and *S. cerevisiae* growth and fermentation. Notably, the antioxidant properties of SLF were enhanced compared with SL. This suggests that substances produced during SL fermentation, such as pigments, can enhance antioxidant capabilities. Subsequently, to examine alterations in VOCs before and after SL fermentation, GC–IMS was employed. A total of 56 VOCs were detected and classified into 12 different compound types. Following the fermentation process, a marked increase in alcohols and esters was observed. These findings can further facilitate the development and design of high-yield MK processes, as well as provide theoretical data for the fermentation of Se-enriched edible fungi food products.

## Figures and Tables

**Figure 1 jof-10-00503-f001:**
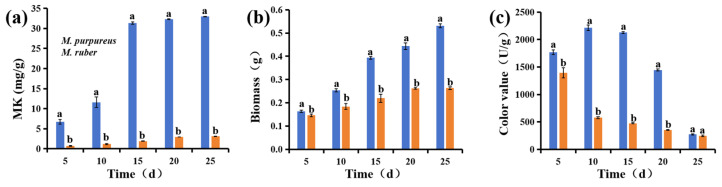
MK content (**a**), biomass (**b**), and color value (**c**) of *M. purpureus* and *M. ruber*. Means with different letters are significantly different at *p* ≤ 0.05.

**Figure 2 jof-10-00503-f002:**
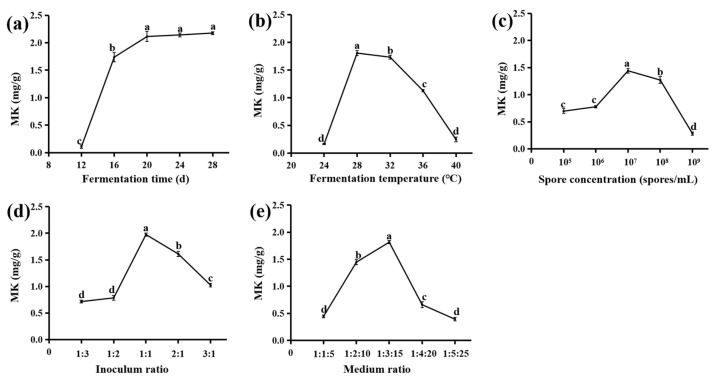
Effect of fermentation time (**a**), fermentation temperature (**b**), spore concentration (**c**), inoculum ratio (**d**), and medium ratio (**e**) on the content of MK. Means with different letters are significantly different at *p* ≤ 0.05.

**Figure 3 jof-10-00503-f003:**
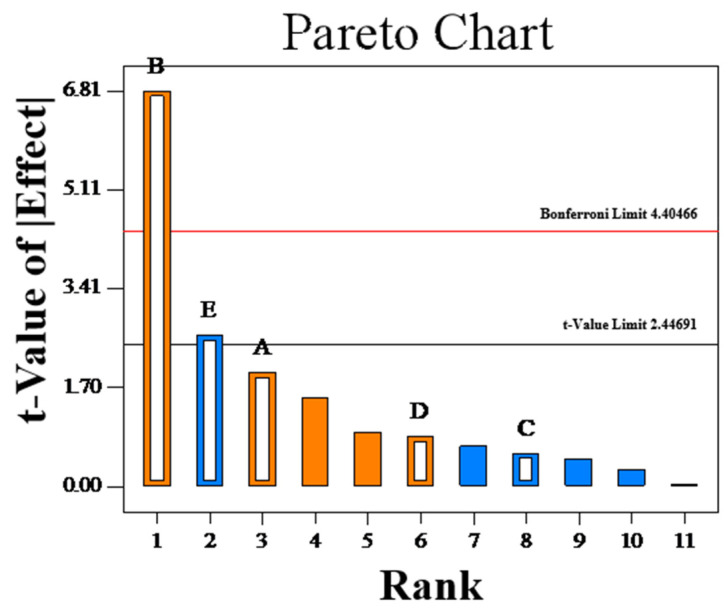
Pareto chart.

**Figure 4 jof-10-00503-f004:**
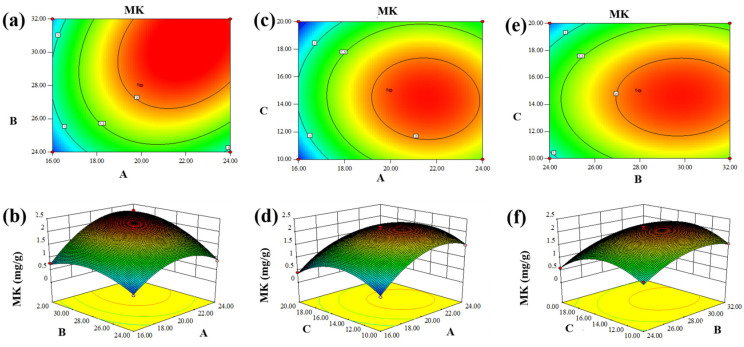
Box-Behnken plots showing the effects of temperature, medium ratio, and fermentation days of fermentation on the MK yield. (**a**,**b**) Effect of the interaction between fermentation time (A) and fermentation temperature (B). (**c**,**d**) Effect of the interaction between fermentation time (A) and medium ratio (C). (**e**,**f**) Effect of the interaction between fermentation temperature (B) and medium ratio (C).

**Figure 5 jof-10-00503-f005:**
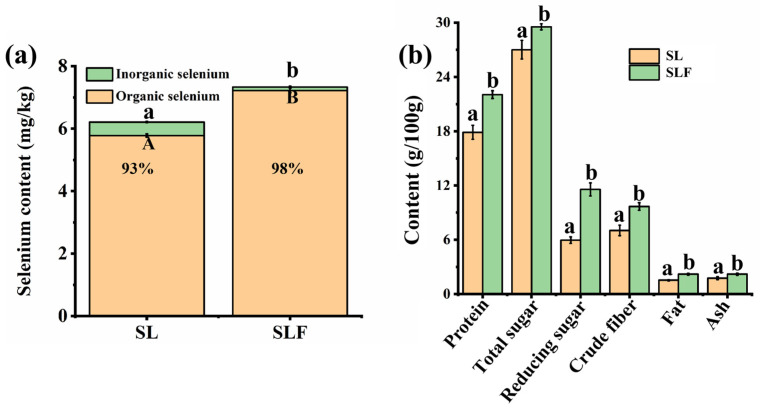
Selenium content (**a**) and nutrient composition (**b**) of SL and SLF. Different lowercase indicate significant differences between total selenium contents of SL and SLF (*p* < 0.05). Different uppercase letters indicate significant differences between organic selenium contents of SL and SLF (*p* < 0.05).

**Figure 6 jof-10-00503-f006:**
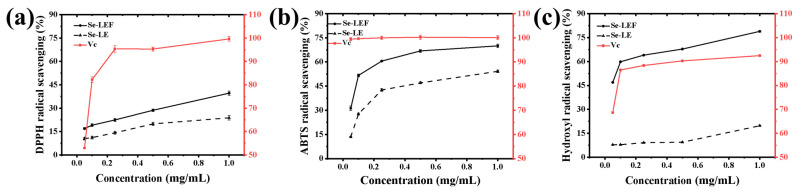
DPPH (**a**), ABTS (**b**), and hydroxyl (**c**) radical scavenging assays of SL and SLF.

**Figure 7 jof-10-00503-f007:**
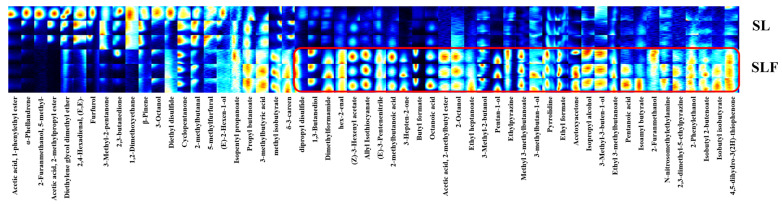
Fingerprint spectra of SL and SLF.

**Table 1 jof-10-00503-t001:** Plackett–Burman experimental scheme design and results.

Test Number	A (Fermentation Time)	B (Fermentation Temperature)	C (Spore Concentration)	D (Medium Ratio)	E (Inoculum Ratio)	Y (Monacolin K mg/g)
1	1	−1	1	1	−1	0.73 ± 0.03
2	1	1	−1	1	1	1.80 ± 0.11
3	−1	1	−1	1	1	1.53 ± 0.02
4	1	−1	−1	−1	1	0.43 ± 0.04
5	1	−1	1	1	1	0.54 ± 0.03
6	−1	−1	1	−1	1	0.28 ± 0.06
7	−1	−1	−1	1	−1	0.45 ± 0.02
8	−1	1	1	−1	1	0.68 ± 0.10
9	−1	1	1	1	−1	1.97 ± 0.08
10	1	1	1	−1	−1	2.09 ± 0.10
11	−1	−1	−1	−1	−1	0.68 ± 0.02
12	1	1	−1	−1	−1	1.99 ± 0.15

**Table 2 jof-10-00503-t002:** Response surface analysis scheme and test results.

Test Number	A (Fermentation Time)	B (Fermentation Temperature)	C (Medium Ratio)	Y (Monacolin K mg/g)
1	−1	0	−1	0.44 ± 0.02
2	1	0	−1	1.49 ± 0.08
3	0	−1	1	0.59 ± 0.02
4	0	0	0	2.05 ± 0.03
5	1	0	1	1.23 ± 0.06
6	1	1	0	2.23 ± 0.11
7	0	1	−1	1.56 ± 0.08
8	0	1	1	1.27 ± 0.08
9	0	−1	−1	0.99 ± 0.03
10	0	0	0	2.21 ± 0.12
11	0	0	0	2.19 ± 0.05
12	1	−1	0	0.87 ± 0.01
13	−1	0	1	0.40 ± 0.16
14	−1	−1	0	0.49 ± 0.02
15	0	0	0	2.12 ± 0.16
16	0	0	0	2.02 ± 0.13
17	−1	1	0	0.77 ± 0.04

**Table 3 jof-10-00503-t003:** Table of variance analysis of the quadratic multinomial model.

Source of Variance	Sum of Square	Degrees of Freedom	Mean Square	F Value	*p*-Value	Significant
Model	7.49	9	0.83	88.49	<0.0001	**
A	1.73	1	1.73	184.39	<0.0001	**
B	1.04	1	1.04	110.68	<0.0001	**
C	0.12	1	0.12	12.89	0.0088	**
AB	0.29	1	0.29	31.11	0.0008	**
AC	0.012	1	0.012	1.30	0.2921	N
BC	0.003	1	0.003	0.37	0.5622	N
A^2^	1.61	1	1.61	171.08	<0.0001	**
B^2^	0.69	1	0.69	73.77	<0.0001	**
C^2^	1.55	1	1.55	164.77	<0.0001	**
Residual	0.066	7	0.009			
Lack of fit	0.038	3	0.013	1.85	0.2779	N
Error term	0.028	4	0.007			
Summation	7.5	16				
R^2^	0.9801					

** Very significant effect, *p* < 0.01; N—insignificant effect, *p* > 0.05.

**Table 4 jof-10-00503-t004:** Amino acid composition of SL and SLF (g/100 g).

Amino Acid	SL	SLF
Asp	2.66 ± 0.01	2.70 ± 0.08
Thr *	0.76 ± 0.02	0.70 ± 0.10
Ser	0.84 ± 0.01	0.57 ± 0.05
Glu	1.59 ± 0.01	1.27 ± 0.01
Pro #	0.07 ± 0.07	0.05 ± 0.01
Gly #	2.34 ± 0.10	3.80 ± 0.11
Ala	0.85 ± 0.01	0.49 ± 0.01
Cys	0.54 ± 0.01	0.37 ± 0.01
Val *	0.71 ± 0.01	0.83 ± 0.01
Met *	1.40 ± 0.08	1.89 ± 0.01
Ile *	0.50 ± 0.01	0.58 ± 0.01
Leu *	0.86 ± 0.02	1.07 ± 0.02
Tyr #	0.24 ± 0.01	0.28 ± 0.02
Phe *	0.71 ± 0.01	1.15 ± 0.01
Lys *	0.75 ± 0.01	2.70 ± 0.17
His	0.26 ± 0.02	0.25 ± 0.02
Arg #	0.91 ± 0.02	0.93 ± 0.02
EAA	5.69 ± 0.33	8.92 ± 0.15
SEAA	3.56 ± 0.30	5.06 ± 0.15
NEAA	6.75 ± 0.10	5.65 ± 0.10
TAA	15.99 ± 0.58	19.63 ± 0.28
EAA/TAA (%)	35.58 ± 0.96	45.44 ± 0.45
NEAA/TAA (%)	42.19 ± 0.64	28.80 ± 0.32
EAA/NEAA (%)	84.33 ± 1.24	179.62 ± 0.59

* Essential amino acids; # semi-essential amino acids.

**Table 5 jof-10-00503-t005:** Gas chromatography–ion mobility spectrometry (GC–IMS ) detection of volatile organic compounds of SL and SLF.

Count	Compound	CAS#	Formula	MW	RI	RT/s	DT/ms
1	Acetic acid, 1-phenylethyl ester	C93925	C_10_H_12_O_2_	164.2	1200.8	992.713	1.8512
2	α-Phellandrene	C99832	C_10_H_16_	136.2	998.3	644.161	1.6989
3	3-Octanol	C589980	C_8_H_18_O	130.2	996.7	640.617	1.7686
4	2-Furanmethanol, 5-methyl-	C3857258	C_6_H_8_O_2_	112.1	961.5	560.070	1.5395
5	Dipropyl disulfide	C629196	C_6_H_14_S_2_	150.3	1097.6	822.724	1.4911
6	Acetic acid, 2-methylpropyl ester	C110190	C_6_H_12_O_2_	116.2	783.6	225.759	1.5749
7	1,3-Butanediol	C107880	C_4_H_10_O_2_	90.1	875.3	364.471	1.3677
8	β-Pinene	C127913	C_10_H_16_	136.2	952.0	537.061	1.2942
9	Diethylene glycol dimethyl ether	C111966	C_6_H_14_O_3_	134.2	956.4	547.544	1.1655
10	2,4-Hexadienal, (E,E)-	C142836	C_6_H_8_O	96.1	915.6	449.702	1.1213
11	Furfurol	C98011	C_5_H_4_O_2_	96.1	827.3	285.468	1.0290
12	Dimethylformamide	C68122	C_3_H_7_NO	73.1	782.3	224.156	1.2600
13	Cyclopentanone	C120923	C_5_H_8_O	84.1	788.8	232.568	1.3445
14	Butyl formate	C592847	C_5_H_10_O_2_	102.1	730.1	159.667	1.5154
15	3-Methyl-2-pentanone	C565617	C_6_H_12_O	100.2	750.5	183.781	1.1630
16	2-Methylbutanal	C96173	C_5_H_10_O	86.1	700.0	127.703	1.1630
17	(E)-3-Pentenenitrile	C16529661	C_5_H_7_N	81.1	694.7	122.656	1.1862
18	1,2-Dimethoxyethane	C110714	C_4_H_10_O_2_	90.1	648.9	86.676	1.3078
19	Diethyl disulfide	C110816	C_4_H_10_S_2_	122.2	919.4	458.547	1.2692
20	5-Methylfurfural	C620020	C_6_H_6_O_2_	110.1	969.1	578.182	1.4177
21	(E)-2-Hexen-1-ol	C928950	C_6_H_12_O	100.2	882.6	378.520	1.2135
22	2,3-Butanedione	C431038	C_4_H_6_O_2_	86.1	584.6	50.460	1.2979
23	Octanoic acid	C124072	C_8_H_16_O_2_	144.2	1199.5	990.506	1.8978
24	Propyl butanoate	C105668	C_7_H_14_O_2_	130.2	890.4	394.096	1.2659
25	Hex-2-enal	C505577	C_6_H_10_O	98.1	855.3	329.170	1.1853
26	(Z)-3-Hexenyl acetate	C3681718	C_8_H_14_O_2_	142.2	1005.6	659.302	1.7491
27	Allyl Isothiocyanate	C57067	C_4_H_5_NS	99.2	887.2	387.570	1.4056
28	2-Methylbutanoic acid	C116530	C_5_H_10_O_2_	102.1	843.1	309.329	1.4541
29	3-Hepten-2-one	C1119444	C_7_H_12_O	112.2	937.6	502.033	1.2445
30	Methyl isobutyrate	C547637	C_5_H_10_O_2_	102.1	668.8	100.688	1.1416
31	Acetic acid, 2-methylbutyl ester	C624419	C_7_H_14_O_2_	130.2	887.8	388.867	1.7504
32	2-Octanol	C123966	C_8_H_18_O	130.2	998.6	644.726	1.4529
33	Isopentyl propanoate	C105680	C_8_H_16_O_2_	144.2	965.1	568.682	1.3516
34	Ethyl heptanoate	C106309	C_9_H_18_O_2_	158.2	1095.2	818.822	1.4096
35	3-Methyl-2-butanol	C598754	C_5_H_12_O	88.1	681.9	111.178	1.2693
36	Pentan-1-ol	C71410	C_5_H_12_O	88.1	767.7	205.242	1.5042
37	Ethylpyrazine	C13925003	C_6_H_8_N_2_	108.1	932.0	488.399	1.0454
38	3-Methylbutyric acid	C503742	C_5_H_10_O_2_	102.1	821.8	277.556	1.2135
39	Methyl 3-methylbutanoate	C556241	C_6_H_12_O_2_	116.2	775.6	215.404	1.1921
40	3-Methylbutan-1-ol	C123513	C_5_H_12_O	88.1	732.8	162.876	1.2340
41	Pyrrolidine	C123751	C_4_H_9_N	71.1	687.5	116.030	1.2241
42	Ethyl formate	C109944	C_3_H_6_O_2_	74.1	617.2	67.894	1.2161
43	Acetoxyacetone	C592201	C_5_H_8_O_3_	116.1	497.0	3.865	1.0386
44	Isopropyl alcohol	C67630	C_3_H_8_O	60.1	510.0	10.823	1.2098
45	3-Methyl-3-buten-1-ol	C763326	C_5_H_10_O	86.1	727.2	156.409	1.5671
46	Ethyl 3-methylbutanoate	C108645	C_7_H_14_O_2_	130.2	840.5	305.328	1.2627
47	Pentanoic acid	C109524	C_5_H_10_O_2_	102.1	899.8	414.112	1.5484
48	Isoamyl butyrate	C106274	C_9_H_18_O_2_	158.2	1054.1	750.099	1.4138
49	δ-3-Carene	C13466789	C_10_H_16_	136.2	1011.0	670.373	1.2420
50	2-Furanmethanol	C98000	C_5_H_6_O_2_	98.1	849.9	320.149	1.1260
51	N-nitrosomethylethylamine	C10595956	C_3_H_8_N_2_O	88.1	832.1	292.625	1.1004
52	2,3-Dimethyl-5-ethylpyrazine	C15707343	C_8_H_12_N_2_	136.2	1093.3	815.589	1.7341
53	2-Phenylethanol	C60128	C_7_H_8_O	122.2	1114.6	850.706	1.2861
54	Isobutyl 2-butenoate	C589662	C_8_H_14_O_2_	142.2	1001.3	650.442	1.3233
55	Isobutyl isobutyrate	C97858	C_8_H_16_O_2_	144.2	902.9	420.756	1.3070
56	4,5-Dihydro-3(2H)-thiophenone	C1003049	C_4_H_6_OS	102.2	941.7	511.871	1.1886

CAS: Chemical Abstracts Service. MW: molecular mass. RI: retention index. RT: retention time. DT: drift time.

## Data Availability

The original contributions presented in the study are included in the article, further inquiries can be directed to the corresponding author.

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
