# Peer review of "Optimization Co-Culture of Monascus purpureus and Saccharomyces cerevisiae on Selenium-Enriched Lentinus edodes for Increased Monacolin K Production"

_jof, 2024, doi:10.3390/jof10070503_

Round 1

Reviewer 1 Report

The manuscript gives the significant contribution to the development of the functional products with added value, based on selenium enriched mushrooms.

Refereeing tables, figures, lines and overall technical part of the manuscript I have no comments and remarks.

The manuscript entitled: “Optimization on selenium-enriched Lentinus edodes 2 fermentation for monacolin K production: nutrient 3 composition, antioxidant properties and flavour analysispresents original research paper that is worth publishing after some major changes.

In section 2.2. Screening of the test strains it is unclear for what purpose screening test was performed? Aditionally, it is unclear how the test was performed, what were the conditions. Were the cultures cultivated and how, was the Lentinus edodes involved in this experiment? What do you mean under "simultaneous culture"? Is this means that both strains are involved at the same time? If so, it should be stated clearly, that both samples are involved at the same time in the fermentation process for the purpose of obtaining monacolin. Additionally, it should be clearly given explanation of how Lentinus was added, was it mycelium or mushroom powder.

What was the selenium concentration in L. edodes? When added to the fermentation medium, what was the concentration of selenium that was achieved?

Line 132-133

Authors stated: The pretest indicated that MK production by composite fermented SL was influenced by several factors, including fermentation temperature, medium ratio, fermentation time, spores concentration, and inoculums ratio.

Reviewer comment: It is unclear how authors obtained these data. It seems that sections 2.2. And 2.3 are connected, but should be better explained and related. Also, please explain what the authors mean by "composite fermented SL".

In section 3.2. It is unclear on the strain of Monascus this experiment refers to.

Line 327-332

Authors stated: The results demonstrated that the optimal MK content is gained when the SL–glucose–distilled water ratio is 1:3:15, reaching 1.81 ± 0.05 mg/g (Figure 2e).  Beyond this ratio, the MK content gradually declines, reaching a minimum concentration of 0.39 ± 0.04 mg/g. As fermentation progressed, the medium lost a significant amount of water, gradually drying out the substrate. This hindered the growth and metabolism of S. cerevisiae, thereby resulting in reduced metabolite production, including MK [24]. Hence, the medium ratio of 1:3:15 is considered optimal.

Reviewer comment: How authors comment declining of MC content that they wrote? What was the main cause of this declining? Also, in this section S. cerevisiae is mentioned, so once again, the very setup of the experiment is veru unclear. It should be given at the beigining, in the section introduction, materials and methods etc…

 In section 3.5. Nutrient composition analysis authors stated that "the organic selenium contents of SL and SLF were 5.78 416 ± 0.05 and 7.22 ± 0.07 mg/kg, which accounted for 93% and 98% of the total selenium".

In material and methods, among determination of nutrient composition (section 2.6.) as method for selenium determination atomic fluorescence spectrometer was given. Using this method organic selenium cannot be determined, only total selenium content. How authors concluded that this is organic selenium?  Additionally, how authors explain and comment their statement that  "the fermentation process was beneficial in increasing the organic selenium content in SL"?

Author Response

Dear reviewer,

We sincerely appreciate you for your careful review and for your professional and thoughtful suggestions. We are sorry for the basic errors in the manuscript. According to your suggestions, we have revised the improper expressions you mentioned. We hope that the revised version will be acceptable for publication in Journal of Fungi. The responses to each point brought up by you are appended in this E-mail. The revised manuscript was uploaded via Submission Center.

Thank you again for your careful review and useful suggestions.

Sincerely yours,

All authors

Response to the first reviewer's questions

  1. In section 2.2. Screening of the test strains it is unclear for what purpose screening test was performed?

Answer: Sorry for our unclarity expression. The screening test was aimed at selecting a strain of Monascus with abundant monacolin K (MK) production, high biomass and excellent color value. MK is an important bioactive substance because of its significant cholesterol-lowering effect; high biomass implies higher yield and enhances the production efficiency; and the color value is a direct reflection of the quality of Monascus pigment. Combining these three aspects, the Monascus purpureus (M. purpureus) will have a wide range of potential applications and market competitiveness. We've added the screening purpose which marked by blue (Line 123-129).

  1. Aditionally, it is unclear how the test was performed, what were the conditions. Were the cultures cultivated and how, was the Lentinus edodes involved in this experiment? What do you mean under "simultaneous culture"? Is this means that both strains are involved at the same time? If so, it should be stated clearly, that both samples are involved at the same time in the fermentation process for the purpose of obtaining monacolin.

Answer: Sorry for our unclarity expression. In the strain screening experiment, Lentinus edodes were not utilized as a culture medium; instead, Potato Dextrose Agar (PDA) was employed. M. purpureus and M. ruber were separately inoculated onto PDA plates overlaid with cellophane. The cultures were collected after 5, 10, 15, 20 and 25 days, freeze-dried and stored in a desiccator for subsequent testing (Line 127-129). During the fermentation process, M. purpureus and Saccharomyces cerevisiae (S. cerevisiae) were both involved to obtain MK. We've added the conditions marked by blue (Line 147).

  1. Additionally, it should be clearly given explanation of how Lentinus was added, was it mycelium or mushroom powder.

Answer: Sorry for our unclarity expression. Selenium-enriched Lentinus edodes powder at 60 mesh, glucose and distilled water were mixed in a certain proportion in a 250 mL conical flask and sterilized at 115°C for 20 min, followed by inoculation of M. purpureus and S. cerevisiae. We've added this information and marked them by blue (Line 112-115).

  1. What was the selenium concentration in edodes? When added to the fermentation medium, what was the concentration of selenium that was achieved?

Answer: The selenium concentration in selenium-enriched Lentinus edodes was 6.21 ± 0.05 mg/kg, and the selenium concentration in selenium-enriched Lentinus edodes fermentation was 7.33 ± 0.05 mg/kg. We've added these data which marked by blue (Line 450-451).

  1. Line 132-133 Authors stated: The pretest indicated that MK production by composite fermented SL was influenced by several factors, including fermentation temperature, medium ratio, fermentation time, spores concentration, and inoculums ratio. Reviewer comment: It is unclear how authors obtained these data. It seems that sections 2.2. And 2.3 are connected, but should be better explained and related. sections 2.2. And 2.3 are connected

Answer: Sorry for our unclarity expression. The objective of the screening test in section 2.2 was aimed at selecting a strain of Monascus with abundant MK production, high biomass and excellent color value. Section 2.3 was involved the co-fermentation of the screened M. purpureus with S. cerevisiae, and explores the effects of fermentation temperature, medium ratio, fermentation time, spores concentration, and inoculum ratio on the production of MK by M. purpureus. We've added it and marked by red (Line 124-127, 147).

  1. Also, please explain what the authors mean by "composite fermented SL". In section 3.2. It is unclear on the strain of Monascus this experiment refers to.

Answer: Thank you for your professional suggestion. Sorry for our unclarity expression. we would like to express that based on the results of the one-way analysis, it was determined that fermentation temperature, medium ratio, fermentation time, spores concentration, and inoculum ratio had a significant effect on the MK yield in the selenium-enriched Lentinus edodes fermentation. We have modified it and marked by blue (Line 155-157).

  1. Authors stated: The results demonstrated that the optimal MK content is gained when the SL–glucose–distilled water ratio is 1:3:15, reaching 1.81 ± 0.05 mg/g (Figure 2e). Beyond this ratio, the MK content gradually declines, reaching a minimum concentration of 0.39 ± 0.04 mg/g. As fermentation progressed, the medium lost a significant amount of water, gradually drying out the substrate. This hindered the growth and metabolism of cerevisiae, thereby resulting in reduced metabolite production, including MK [24]. Hence, the medium ratio of 1:3:15 is considered optimal. Reviewer comment: How authors comment declining of MC content that they wrote? What was the main cause of this declining?

Answer: Thank you for your professional suggestion. The primary reason for the decline in MK production may lie in the gradual depletion of nutrients in the fermentation medium over time. Specifically, as glucose and other carbon sources diminish, the metabolic activities of microorganisms are impacted, leading to reduced synthesis of MK and other metabolic products. Furthermore, studies have shown that S. cerevisiae can stimulate the production of secondary metabolites in M. purpureus [1]. However, when nutrients in the medium become scarce and the environment turns unfavorable, the growth rate of yeast may slow down or even cease, accompanied by a corresponding decrease in metabolic activities, ultimately resulting in reduced MK synthesis. We have modified it marked by blue (Line 360-364).

  1. Moghadam, H. D.; Tabatabaee Yazdi, F.; Shahidi, F.; Sarabi-Jamab, M.; Es'haghi, Z. Co-culture of Monascus purpureus with Saccharomyces cerevisiae: A strategy for pigments increment and citrinin reduction. Biocatal Agr Biotech. 2022, 45, 102501. https://doi.org/10.1016/j.bcab.2022.102501.

  1. Also, in this section cerevisiae is mentioned, so once again, the very setup of the experiment is veru unclear. It should be given at the beigining, in the section introduction, materials and methods etc…

Answer: Sorry for our unclarity expression. I have supplemented these information in the introduction, materials and methods sections which are marked by blue (Line 71-81, 109, 147-148).

  1. In section 3.5. Nutrient composition analysis authors stated that "the organic selenium contents of SL and SLF were 5.78 ± 0.05 and 7.22 ± 0.07 mg/kg, which accounted for 93% and 98% of the total selenium". In material and methods, among determination of nutrient composition (section 2.6.) as method for selenium determination atomic fluorescence spectrometer was given. Using this method organic selenium cannot be determined, only total selenium content. How authors concluded that this is organic selenium?

Answer: Sorry for our unclarity expression. Atomic fluorescence spectrometry allowed the determination of total selenium content and inorganic selenium content, followed by differential subtraction to obtain the organic selenium content. The differential subtraction method was performed according to Zhou et al [2]. We have added the methods for determining inorganic selenium in the manuscript and marked by blue (Line 178-191, 195-197).

  1. Zhou, Q.; Lei, M.; Li, J.; Wang, M.; Zhao, D.; Xing, A.; Zhao, K. Selenium speciation in tea by dispersive liquid–liquid microextraction coupled to high‐performance liquid chromatography after derivatization with 2,3‐diaminonaphthalene. J Sep Sci. 2015, 38, 1577-1583. https://doi.org/10.1002/jssc.201401373.

  1. Additionally, how authors explain and comment their statement that "the fermentation process was beneficial in increasing the organic selenium content in SL"?

Answer: What we want to state is that the inorganic selenium in selenium-enriched Lentinus edodes (SL) is converted to organic selenium by S. cerevisiae and M. purpureus through fermentation. This may be attributed to the fact that S. cerevisiae and M. purpureus use their own metabolic pathways and enzyme systems to absorb inorganic selenium (e.g., selenate, selenite) from the culture medium or substrate into the cells during the fermentation process, and convert inorganic selenium into organic selenium forms such as selenoamino acids, selenoproteins, selenopolysaccharides, and other organic selenium forms through processes such as reduction, methylation, and selenoprotein synthesis. These organic selenium forms are not only more stable, but also more easily absorbed and utilized by the human body or other organisms. We've added this explanation and marked by blue (Line 450-459).

Reviewer 2 Report

Comments for Authors

Dear Authors

Firstly, I congratulate the authors on their manuscript. My main comments and suggestions, which hopefully can help the author(s) improve the study. My recommendation is „Minor revision”.

Sincerely,

Your Reviewer

Comments and suggestions

Abstract

The abstract is well-structured, but I would like to add a small extension in the case of shiitake, as Se-enriched shiitake is not “just” an edible mushroom species, but in this case a cultivated strain. In the results presented, it is not clear what the values are higher than for the different parameters tested, e.g.: Monacolin K content is produced in higher amounts during fermentation compared to M. purpureus. It is also not clear whether L. edodes was the fermenting agent or S. cerevisiae.

Keywords

In my opinion, some keywords are not focused enough, I would definitely include monacolin K and antioxidant because they are as a large part of the manuscript is organised around this.

Introduction

In my opinion, the introduction is a little longer than usual. But all parts are related to the subject of the research.

Materials and Methods

In this section, it was surprising to see two different strains of M. purpureus and M. ruber., because you did not mention in the abstract or introduction.

While S. cerevisiae is not indicated in this subsection, nor is the strain code.

I would rewrite subsection 2.1, as several strains and a few chemicals appear, while the other substances appear elsewhere. This could be clarified either by a new title or a major expansion.

The title of subsection 2.2 is also inappropriate, as it is mostly about sample preparation and MK determination. I could imagine the latter in subchapter 2.3, as this is where the parameters that affect MK production are investigated, so this would be the best place to discuss the method to detect MK in the first place. S. cerevisiae appears here, but not before, so its role is not clear.

After subsection 2.6, I see no point in further subsections, especially as one of them has no more than two lines. A better solution would be to have a table of methods of composition, which would include the group of the materials, the name of the method and the reference. Then to text the amendments, so that this section would be more compact.

The sample preparation of antioxidant methods should be moved to the sample preparation section earlier, and the presentation of other methods for the same sample should be moved to this section.

Results and Analysis

In the case of the results, I am thinking mainly of Fig 2., I do not recognise the individual parallel treatment.

I mean, a duration of fermentation has a temperature, a certain spore concentration, and a given inoculum ratio. But I don't see the other data, for example, how changes the amount of MK during the fermentation with different spore concentrations varied on different temperature.

I suppose one spore concentration involved incubating at five different temperatures. Or if it is not I didn't see the control into any results, so I can't decide which combination of treatments is the most optimal. Here some sort of three-dimensional surface diagram would be most useful, so that three different data could be validated simultaneously and the joint effect of the different parameters could be detected, which is done into Fig. 4.. But there is a question, why is Fig. 2 needed? It is not clear whether M. purpureus and M. ruber or S. cerevisiae are inoculated together.

A Fig. 3 and the text is not clear, especially as five different factors are examined by the authors, while Fig. 3 shows eleven data points. A Fig. 5 a) is not informative, this should simply be written down in the text of this result.

The yield of MK described in the conclusions (2.42 mg/g) is not clear for myself, because based on Fig. 2 the MK content was 32.97 mg/g. Please resolve this discrepancy for me.

Comments for Authors

Dear Authors

Firstly, I congratulate the authors on their manuscript. My main comments and suggestions, which hopefully can help the author(s) improve the study. My recommendation is „Minor revision”.

Sincerely,

Your Reviewer

Comments and suggestions

Abstract

The abstract is well-structured, but I would like to add a small extension in the case of shiitake, as Se-enriched shiitake is not “just” an edible mushroom species, but in this case a cultivated strain. In the results presented, it is not clear what the values are higher than for the different parameters tested, e.g.: Monacolin K content is produced in higher amounts during fermentation compared to M. purpureus. It is also not clear whether L. edodes was the fermenting agent or S. cerevisiae.

Keywords

In my opinion, some keywords are not focused enough, I would definitely include monacolin K and antioxidant because they are as a large part of the manuscript is organised around this.

Introduction

In my opinion, the introduction is a little longer than usual. But all parts are related to the subject of the research.

Materials and Methods

In this section, it was surprising to see two different strains of M. purpureus and M. ruber., because you did not mention in the abstract or introduction.

While S. cerevisiae is not indicated in this subsection, nor is the strain code.

I would rewrite subsection 2.1, as several strains and a few chemicals appear, while the other substances appear elsewhere. This could be clarified either by a new title or a major expansion.

The title of subsection 2.2 is also inappropriate, as it is mostly about sample preparation and MK determination. I could imagine the latter in subchapter 2.3, as this is where the parameters that affect MK production are investigated, so this would be the best place to discuss the method to detect MK in the first place. S. cerevisiae appears here, but not before, so its role is not clear.

After subsection 2.6, I see no point in further subsections, especially as one of them has no more than two lines. A better solution would be to have a table of methods of composition, which would include the group of the materials, the name of the method and the reference. Then to text the amendments, so that this section would be more compact.

The sample preparation of antioxidant methods should be moved to the sample preparation section earlier, and the presentation of other methods for the same sample should be moved to this section.

Results and Analysis

In the case of the results, I am thinking mainly of Fig 2., I do not recognise the individual parallel treatment.

I mean, a duration of fermentation has a temperature, a certain spore concentration, and a given inoculum ratio. But I don't see the other data, for example, how changes the amount of MK during the fermentation with different spore concentrations varied on different temperature.

I suppose one spore concentration involved incubating at five different temperatures. Or if it is not I didn't see the control into any results, so I can't decide which combination of treatments is the most optimal. Here some sort of three-dimensional surface diagram would be most useful, so that three different data could be validated simultaneously and the joint effect of the different parameters could be detected, which is done into Fig. 4.. But there is a question, why is Fig. 2 needed? It is not clear whether M. purpureus and M. ruber or S. cerevisiae are inoculated together.

A Fig. 3 and the text is not clear, especially as five different factors are examined by the authors, while Fig. 3 shows eleven data points. A Fig. 5 a) is not informative, this should simply be written down in the text of this result.

The yield of MK described in the conclusions (2.42 mg/g) is not clear for myself, because based on Fig. 2 the MK content was 32.97 mg/g. Please resolve this discrepancy for me.

Author Response

Dear reviewer,

We sincerely appreciate you for your careful review and for your professional and thoughtful suggestions. We are sorry for the basic errors in the manuscript. According to your suggestions, we have revised the improper expressions you mentioned. We hope that the revised version will be acceptable for publication in Journal of Fungi. The responses to each point brought up by you are appended in this E-mail. The revised manuscript was uploaded via Submission Center.

Thank you again for your careful review and useful suggestions.

Sincerely yours,

All authors

Response to the second reviewer's questions

  1. The abstract is well-structured, but I would like to add a small extension in the case of shiitake, as Se-enriched shiitake is not “just” an edible mushroom species, but in this case a cultivated strain. In the results presented, it is not clear what the values are higher than for the different parameters tested, e.g.: Monacolin K content is produced in higher amounts during fermentation compared to purpureus. It is also not clear whether L. edodes was the fermenting agent or S. cerevisiae.

Answer: Thank you for your professional suggestion. Sorry for our unclarity expression. Selenium-enriched L. edodes powder is the fermentation substrate, and M. purpureus and S. cerevisiae are the fermentation strains. We've added these information which marked by red (Line 147).

  1. In my opinion, some keywords are not focused enough, I would definitely include monacolin K and antioxidant because they are as a large part of the manuscript is organised around this.

Answer: Thank you for your professional suggestion. We've included monacolin K and antioxidant as keywords and marked by purple (Line 37-38).

  1. In this section, it was surprising to see two different strains of purpureus and M. ruber., because you did not mention in the abstract or introduction.

Answer: Thank you for your professional suggestion. We've added the information about M. purpureus and M. ruber in the introduction. (Line 60-64).

  1. While cerevisiae is not indicated in this subsection, nor is the strain code.

Answer: Thank you for your professional suggestion. We've added strain code for S. cerevisiae marked by purple (Line 109).

  1. I would rewrite subsection 2.1, as several strains and a few chemicals appear, while the other substances appear elsewhere. This could be clarified either by a new title or a major expansion.

Answer: Thank you for your professional suggestion. We have expanded subsection 2.1 by adding the preparation of fermentation media and marked them by blue (Line 112-118).

  1. The title of subsection 2.2 is also inappropriate, as it is mostly about sample preparation and MK determination. I could imagine the latter in subchapter 2.3, as this is where the parameters that affect MK production are investigated, so this would be the best place to discuss the method to detect MK in the first place. cerevisiae appears here, but not before, so its role is not clear.

Answer: Thank you for your professional suggestion. The section 2.2 is been renamed as "Determination of MK, biomass and color values for strain screening," and within this section, a comprehensive description of the microbial strain cultivation process has been provided and marked by blue (Line 123-129). In the introduction we described that the related literature shows that fermentation of S. cerevisiae and Monascus is favourable to promote the production of secondary metabolite aspects of Monascus such as MK, so we are using S. cerevisiae and M. purpureus in a co-fermentation (Line 71-81). We have added S. cerevisiae co-fermentation strain code to the material marked by blue (Line 109).

  1. After subsection 2.6, I see no point in further subsections, especially as one of them has no more than two lines. A better solution would be to have a table of methods of composition, which would include the group of the materials, the name of the method and the reference. Then to text the amendments, so that this section would be more compact.

Answer: Thank you for your professional suggestion. We combined sections 2.6.3 and 2.6.2 into a single chapter, but due to the extensive content of 2.6.1, which covers the determination of total selenium and inorganic selenium, we opted not to merge this section. We have modified and marked them by purple (Line 199-206)

  1. The sample preparation of antioxidant methods should be moved to the sample preparation section earlier, and the presentation of other methods for the same sample should be moved to this section.

Answer: Thank you for your professional suggestion. Actually, the sample preparation method specifically designed for antioxidant experiments is solely suitable for such use, and therefore, it has been deliberately placed within the antioxidant testing section.

  1. In the case of the results, I am thinking mainly of Fig 2., I do not recognise the individual parallel treatment. I mean, a duration of fermentation has a temperature, a certain spore concentration, and a given inoculum ratio. But I don't see the other data, for example, how changes the amount of MK during the fermentation with different spore concentrations varied on different temperature. I suppose one spore concentration involved incubating at five different temperatures.

Answer: Thank you for your professional suggestion. Through preliminary experiments, we identified the five factors that have the greatest impact on the MK content in selenium-rich Lentinus edodes fermentation. Based on relevant literature, it is generally believed that the optimal growth temperature for the commonly used fermentation strains, M. purpureus and S. cerevisiae, ranges from 25 to 30°C. Therefore, we did not include the factor of a single spore concentration incubated at five different temperatures as an influencing variable.

  1. Or if it is not I didn't see the control into any results, so I can't decide which combination of treatments is the most optimal. Here some sort of three-dimensional surface diagram would be most useful, so that three different data could be validated simultaneously and the joint effect of the different parameters could be detected, which is done into Fig. 4.. But there is a question, why is Fig. 2 needed?

Answer: Thank you for your professional suggestion. Fig. 2, which presents the results of a single-factor experiment, serves to preliminarily identify the factors that have a significant impact on the MK content in selenium-rich Lentinus edodes fermentation. This process allows for the screening of the most crucial factors, thereby providing a foundation for subsequent response surface methodology (Fig. 4) analysis.

  1. It is not clear whether purpureus and M. ruber or S. cerevisiae are inoculated together.

Answer: Sorry for our unclarity expression. The experiment involved simultaneously inoculating M. purpureus and S. cerevisiae to co-ferment selenium-rich Lentinus edodes. In a 250 mL conical flask, 2 g of selenium-enriched Lentinus edodes powder were dispensed, followed by the addition of glucose and distilled water in a specified ratio. The mixture was then sterilized at 115°C for 20 minutes, after which it was inoculated with M. purpureus and S. cerevisiae. Monascus is surface cultured. We've added these information and marked them by blue (Line 115-118, 147).

  1. A Fig. 3 and the text is not clear, especially as five different factors are examined by the authors, while Fig. 3 shows eleven data points. A Fig. 5 a) is not informative, this should simply be written down in the text of this result.

Answer: Thank you for your professional suggestion. Figure 3 illustrates the individual effects of five factors on the production of MK in fermentation experiments. The horizontal axis of the Pareto chart represents different factors, encompassing the five factors investigated in the single-factor experiments, while also integrating the collective influence of all factors. Notably, the numerical value 11 on the horizontal axis is unrelated to the number of experimental trials.

  1. The yield of MK described in the conclusions (42 mg/g) is not clear for myself, because based on Fig. 2 the MK content was 32.97 mg/g. Please resolve this discrepancy for me.

Answer: Thank you for your professional suggestion. The likely reason is that the medium used for screening the M. purpureus was PDA (32.97 mg/g MK), whereas the fermentation medium employed the selenium-rich Lentinus edodes powder as the fermentation substrate, supplemented with a certain proportion of glucose and distilled water (2.42 mg/g). So the MK contents were different mainly because the culturing medium is different.

Reviewer 3 Report

From the materials and methods section it is not entirely clear how the fermentation process was carried out.

2.2 -It is necessary to indicate which flasks were cultured in and what volume of medium. Specify the exact composition of the medium. Was it submerged  or surface cultivation of Monascus?

L117- under specific conditions - Which ones?

Section 2.2 should be renamed as a definition of MK and separated into a separate one. Before this section, make a section on strains cultivation. Describe in detail.

L 135 - Different spores concentrations- What fungus spores?

L282 - of fungus- Which one?

L305 - . The findings suggest that a temperature of 28°C is the optimal growth temperature for both M. purpureus and S. cerevisiae- No data was obtained about S. cerevisiae.

L 327 - as the substrate matrix - How was it received? Specify in methods.

The title  should be changed to something like this

Optimization Co-culture of Monascus purpureus and  Saccharomyces cerevisiae on selenium-enriched Lentinus edodes for  increase monacolin K production.

In the abstract, percentage values ​​should be rounded to whole numbers.

From the materials and methods section it is not entirely clear how the fermentation process was carried out.

2.2 -It is necessary to indicate which flasks were cultured in and what volume of medium. Specify the exact composition of the medium. Was it submerged  or surface cultivation of Monascus?

L117- under specific conditions - Which ones?

Section 2.2 should be renamed as a definition of MK and separated into a separate one. Before this section, make a section on strain cultivation. Describe in detail.

L 135 - Different spores concentrations- What fungus spores?

L282 - of fungus- Which one?

L305 - . The findings suggest that a temperature of 28°C is the optimal growth temperature for both M. purpureus and S. cerevisiae- No data was obtained about S. cerevisiae.

L 327 - as the substrate matrix - How was it received? Specify in methods.

The title  should be changed to something like this-

Optimization Co-culture of Monascus purpureus and  Saccharomyces cerevisiae on selenium-enriched Lentinus edodes for  increase monacolin K production.

In the abstract, percentage values ​​should be rounded to whole numbers.

Author Response

Dear reviewer,

We sincerely appreciate you for your careful review and for your professional and thoughtful suggestions. We are sorry for the basic errors in the manuscript. According to your suggestions, we have revised the improper expressions you mentioned. We hope that the revised version will be acceptable for publication in Journal of Fungi. The responses to each point brought up by you are appended in this E-mail. The revised manuscript was uploaded via Submission Center.

Thank you again for your careful review and useful suggestions.

Sincerely yours,

All authors

Response to the third reviewer's questions

  1. From the materials and methods section it is not entirely clear how the fermentation process was carried out. 2.2 -It is necessary to indicate which flasks were cultured in and what volume of medium. Specify the exact composition of the medium. Was it submerged or surface cultivation of Monascus?

Answer: Thank you for your professional suggestion. In a 250 mL conical flask, 2 g of selenium-enriched Lentinus edodes powder were dispensed, followed by the addition of glucose and distilled water in a specified ratio. The mixture was then sterilized at 115°C for 20 minutes, after which it was inoculated with M. purpureus and S. cerevisiae. Monascus is surface cultured. We've added these information and marked them by blue (Line 115-118).

  1. L117- under specific conditions - Which ones?

Answer: Thank you for your professional suggestion. The specific conditions is M. purpureus and M. ruber were inoculated onto Potato Dextrose Agar (PDA) plates overlaid with cellophane. We've added it which marked by blue (Line 127-128).

  1. Section 2.2 should be renamed as a definition of MK and separated into a separate one. Before this section, make a section on strains cultivation. Describe in detail.

Answer: Thank you for your professional suggestion. The section 2.2 is been renamed as "Determination of MK, biomass and color values for strain screening," and within this section, a comprehensive description of the microbial strain cultivation process has been provided and marked by blue (Line 123-129).

  1. L 135 - Different spores concentrations- What fungus spores?

Answer: Thank you for your professional suggestion. The different spore concentrations are those of M. purpureus and S. cerevisiae. We've added it marked by red (Line 145).

  1. L282 - of fungus- Which one?

Answer: Thank you for your professional suggestion. This fungus refers to the M. purpureus and S. cerevisiae. We have modified it marked by red (Line 307-310).

  1. L305 - . The findings suggest that a temperature of 28°C is the optimal growth temperature for both purpureus and S. cerevisiae- No data was obtained about S. cerevisiae.

Answer: Sorry for our unclarity expression. We can’t conclude from the experimental results that 28°C is the optimal growth temperature for both M. purpureus and S. cerevisiae. Rather, the correct interpretation is that at 28°C, the MK content in SLF is maximized. It is widely recognized that the optimum growth temperature of S. cerevisiae, as a commonly used fermentation strain, is generally between 28 and 30℃ [1]. Although the specific data of S. cerevisiae at 28°C were not explicitly stated, S. cerevisiae was able to synergistically ferment M. purpureus with a view to increasing the MK-producing capacity during selenium-enriched Lentinus edodes fermentation, and so we chose 28°C as the optimal temperature for fermentation. We have modified it and marked by red (Line 331-332).

  1. Zhu, L.; Zhang, X.; Wang, Y.; Gao, X.; Xu, Q.; Liu, W.; Xu, Q.; Zhao, D.; Cai, J. Recovery and characterization of β-glucosidase-producing non-Saccharomyces yeasts from the fermentation broth of Vitis labruscana Baily × Vitis vinifera L. for investigation of their fermentation characteristics. Arch Microbiol. 2024, 206, 174. https://doi.org/10.1007/s00203-024-03878-9.

  1. L 327 - as the substrate matrix - How was it received? Specify in methods.

Answer: Thank you for your professional suggestion. In a 250 mL conical flask, 2 g of selenium-enriched Lentinus edodes powder were dispensed, followed by the addition of glucose and distilled water in a specified ratio. The mixture was then sterilized at 115°C for 20 minutes, after which it was inoculated with M. purpureus and S. cerevisiae. We've added these information and marked them by blue (Line 115-118).

  1. The title should be changed to something like this Optimization Co-culture of Monascus purpureus and Saccharomyces cerevisiae on selenium-enriched Lentinus edodes for increase monacolin K production.

Answer: Thank you for your professional suggestion. We have modified the title as you suggested (Line 2-4).

  1. In the abstract, percentage values should be rounded to whole numbers.

Answer: Thank you for your professional suggestion. We have rounded the percentage values to whole numbers and marked them by red (Line 31-33).